

# A large-$N$ approach to magnetic impurities in superconductors

**Chen-How Huang[1,2], Alejandro M. Lobos[3,4] and Miguel A. Cazalilla[1,5]**

**1** Donostia International Physics Center (DIPC), Manuel de Lardizábal 4,
20018 San Sebastián, Spain

**2** Departamento de Polímeros y Materiales Avanzados: Física, Química y Tecnología,
Facultad de Ciencias Químicas, Universidad del País Vasco UPV/EHU,
20018 Donostia-San Sebastián, Spain

**3** Facultad de Ciencias Exactas y Naturales (UNCuyo),
Padre Jorge Contreras 1300, CP 5500, Mendoza, Argentina

**4** Instituto Interdisciplinario de Ciencias Básicas (CONICET-UNCuyo),
Padre Jorge Contreras 1300, CP 5500, Mendoza, Argentina

**5** IKERBASQUE, Basque Foundation for Science,
Plaza Euskadi 5 48009 Bilbao, Spain

## Abstract

Quantum spin impurities coupled to superconductors are under intense investigation for their relevance to fundamental research as well as the prospects to engineer novel quantum phases of matter. Here we develop a large-$N$ mean-field theory of a strongly coupled spin-$\frac{1}{2}$ quantum impurity in a conventional $s$-wave superconductor. The approach is benchmarked against Wilson's numerical renormalization group (NRG). While the large-$N$ method is not applicable in the weak-coupling regime where the Kondo temperature $T_K$ is smaller than the superconducting gap $\Delta$, it performs very well in the strong coupling regime where $T_K \gtrsim \Delta$, thus allowing us to obtain a reasonably accurate description of experimentally relevant quantities. The latter include the energy of the Yu-Shiba-Rusinov subgap states, their spectral weight, as well as the local density of continuum states. The method provides a reliable analytical tool that complements other perturbative and non-perturbative methods, and can be extended to more complex impurity models for which NRG may not be easily applicable.

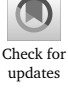
doi:10.21468/SciPostPhys.18.3.087

# 1 Introduction

Magnetic impurities adsorbed on the surface of superconductors are a unique platform to study the competition between superconductivity and magnetism at the atomic scale [1, 2]. Low-temperature scanning-tunneling microscopy (STM) techniques allow to address spectral and real-space properties of adsorbate-surface nanostructures with unprecedented precision. The differential conductance measured using STM near the impurity reveals the presence of Yu-Shiba-Rusinov (YSR) states, which emerge due to the disruption of the superconducting state introduced by the local exchange field of the impurity. These states were originally predicted in the seminal work of Yu [3], Shiba [4], and Rusinov [5], who assumed that the quasi-particle scattering with the magnetic impurity can be treated as a classical local magnetic field. YSR states appear as resonances in the differential conductance, symmetrically located around the Fermi level at energies within the superconducting gap $\Delta$, and are spatially localized around the impurity [6]. As a consequence of the interplay between quantum fluctuations, Kondo screening, single-ion anisotropy, etc., YSR states can display a very complex behavior, as revealed in recent experimental STM works [7–14].

From a theoretical perspective, a full quantum treatment of the complex behavior of a magnetic impurity in a superconductor leads to a many-body problem in which Kondo and pairing correlations compete with each other [1, 15–19]. For the case of a quantum spin-$\frac{1}{2}$ impurity coupled to a $s$-wave superconductor via a weak exchange coupling $J$, pairing correlations favor a doublet ground state in which the quantum impurity effectively decouples from the superconductor. On the other hand, for sufficiently large $J$, and hence large Kondo temperature $T_K$, a Kondo singlet with a fully screened impurity spin is favored. Using a generalization of Wilson's [20] numerical renormalization group (NRG) [21–23] and more recently, the density-matrix renormalization group (DMRG) [24] techniques, the doublet-singlet quantum phase transition has been shown to occur for $T_K \approx \Delta$, where $\Delta$ is the pairing gap. This transition is evidenced in the spectral properties by the energy crossing of the two symmetrically located YSR levels, which can be directly seen in the STM differential conductance signal [10, 25].

The "classical" approach of YSR, which is often uncritically employed to deal with magnetic impurities in superconductors, nonetheless successfully captures several important features of the full quantum problem, such as the level-crossing transition. However, this approach also suffers from a number of drawbacks, beginning with the prediction that the level crossing takes place for an unreasonably large value of $J$. Furthermore, it also assigns definite spin quantum

number to the YSR excitations [3–5], which is particularly inaccurate in the strong coupling regime. These drawbacks stem from the complete neglect of quantum fluctuations of the local magnetic moment. Recently, quantum fluctuation effects have been studied within a "single-site" approximation [26,27], which, by a rather severe truncation of the Hilbert space, renders the problem tractable using modest numerics and, in some cases, even analytical methods. However, the results of this kind of approach can be at best regarded as qualitative.

In this article, motivated by the theoretical challenges described above, we investigate a large-$N$ approach to an SU($N$) extension of the impurity problem in a superconductor. The same approach correctly captures the formation of the Kondo singlet in normal metals [28–30], and here we show it is surprisingly accurate in describing the competition between the latter and superconducting pairing correlations for a spin-1/2 impurity in the strong coupling regime. It is worth noticing that other large-$N$ approaches have been also deployed to tackle this problem [31,32]. In particular, in Ref. [31] a diagrammatic approach called "non-crossing approximation" (NCA) was applied to a large-$N$ generalization of the problem described by the Anderson model with infinite onsite Coulomb repulsion. Within the NCA, the resulting set of self-consistent integral equations are numerically solved, which allows to access the spectral properties. However, besides being technically challenging, it is also known that in normal metals the NCA fails to correctly describe Kondo correlations at $T \lesssim T_K$ [33].

In Ref. [32] the saddle-point approximation was applied to a generalization of the impurity model with SU($N$)-symmetry. By computing the free energy as a function of the impurity magnetization, it was shown that this method always yields a Kondo singlet as the ground state in any parameter regime and therefore it does not capture the level-crossing transition. Specifically, in Ref. [32] an $N$-orbital model with SU($N$)-orbital symmetry that keeps intact the SU(2)-spin symmetry of the original model is studied. In contrast, here we study a model that is mathematically equivalent to a Kondo impurity in a superconductor by extending its SU(2)-spin symmetry to SU($N$), and we show that the transition to the Kondo singlet can be captured within the saddle-point approximation. However, this saddle-point cannot describe the spectral properties in the weak coupling regime where the magnetic moment remains unscreened. Nevertheless, in the strong coupling regime, we show by carefully comparing our results to those obtained using NRG for the SU(2)-symmetric model that the spectral properties are fairly well described. In particular, we find the saddle-point approximation reproduces well both the location of the transition and position of the YSR in the Kondo screened phase. These results are encouraging and indicate that the present large-$N$ approach may be a framework that is both conceptually and technically simple and capable of describing the strong coupling regime. Indeed, its technical complexity is just a bit higher than the large spin-$S$ "classical approximation" pioneered by Yu, Shiba, and Rusinov [3–5], as the mean-field Hamiltonian is quadratic and the mean-field parameters must be obtained by solving a set of nonlinear self-consistent equations. The availability of this well-tested approach opens the possibility of using it to compute both spectral and real-space properties of complex systems such as magnetic chains or lattices of various geometries [34, 35] as well as superconducting hetero-structures [36] in the strong coupling regime. For the latter, NRG or similarly accurate but numerically-intensive numerical methods may not be easily applicable.

The rest of this article is organized as follows: In Sec. 2 we introduce the theoretical model, discuss its extension to SU($N$) symmetry, and derive the saddle-point equations in the $N \to \infty$ limit. In Sec. 3, we compare to NRG the results of our large $N$ approach for the position of the YSR excitations, their spectral weight, and the spectral density of continuum states. Finally, in Sec. 4 we provide our conclusions. The Appendices contain important details and some generalizations of the calculations and methods.

## 2 Model and SU($N$) generalization

We start with the following model of a spin-$\frac{1}{2}$ magnetic impurity coupled to a conventional $s$-wave superconductor:

$$H = H_c + H_{\text{imp}}, \tag{1}$$

$$H_c = \sum_{\vec{k},\sigma} \xi_{\vec{k}} d^\dagger_{\vec{k},\sigma} d_{\vec{k},\sigma} + \Delta \sum_{\vec{k}} \left[ d_{\vec{k},\uparrow} d_{-\vec{k},\downarrow} + \text{H.c.} \right], \tag{2}$$

$$H_{\text{imp}} = J \vec{S} \cdot \vec{s}_0, \tag{3}$$

where $H_c$ describes the superconductor electronic degrees of freedom, represented by the operators $d^\dagger_{\vec{k},\sigma}$ ($d_{\vec{k},\sigma}$) which create (destroy) an electron with lattice wave vector $\vec{k}$ and spin $\sigma = \uparrow, \downarrow$. The band dispersion $\xi_{\vec{k}} = \epsilon_{\vec{k}} - \mu$ is referred to the chemical potential $\mu$. The pairing potential $\propto \Delta$ describes, within the BCS mean-field approximation, the superconducting correlations between the electrons. The term $H_{\text{imp}}$ is the Kondo (or $s$-$d$) exchange coupling between the superconductor and the local magnetic moment of the impurity described by the SU(2) spin operator $\vec{S} = (S^x, S^y, S^z)$. The local electron spin operator at the origin is defined as $\vec{s}_0 = \frac{1}{2} \sum_{\sigma,\sigma'} d^\dagger_{0,\sigma} \boldsymbol{\sigma}_{\sigma\sigma'} d_{0,\sigma'}$, with $d_{0,\sigma} = \sum_{\vec{k}} d_{\vec{k},\sigma}/\sqrt{\Omega}$, where $\Omega$ is the system volume and $\boldsymbol{\sigma} = (\sigma^x, \sigma^y, \sigma^z)$ the vector of spin Pauli matrices. We shall further assume the coupling to the impurity contains no scattering potential. This is a reasonable assumption if the density of states of the host in the normal state (i.e. for $\Delta = 0$) and the hybridization of the magnetic impurity level are well approximated by constants over wide energy range (typically larger than the characteristic energy scales of the magnetic impurity), as it is often the case for many kinds of metals and magnetic impurities. Under such conditions (referred to below as the "wide band" limit) particle-hole symmetry is realized. As discussed below, this additional symmetry plays an important role in the extension of the above Hamiltonian to a fully SU($N$)-symmetric model.

While not necessary for the large-$N$ method, we can simplify the theoretical treatment and subsequent calculations by assuming an isotropic metal (i.e., the jellium model) and exploit the spherical symmetry. Expanding the Bloch waves $\vec{k}$ in a spherical waves, it can be shown that the magnetic impurity couples only to the $s$-wave scattering channel, and therefore the spatial dimensionality of the problem effectively reduces to the radial coordinate [30, 33]. A phenomenological model representing this effective one-dimensional problem corresponds to a semi-infinite tight-binding chain with the impurity spin $\vec{S}$ coupled to the leftmost site at $j = 0$. This derivation is similar in spirit, albeit not strictly equivalent, to the Wilson chain implemented in the NRG method [20,21,33,37]. Note that this "Wilson-like" chain is just a toy-model Hamiltonian that captures the main features of the generic bulk Hamiltonian Eq. (2), with the advantage of being much more tractable and amenable for analytical calculations. However, we stress that this simplification is not essential and does not change our results qualitatively. The original bandstructure in Eq. (2) can be used whenever necessary, as long as it is particle-hole symmetric.

The simplified one-dimensional model therefore reads:

$$H = H_c + H_{\text{imp}}, \tag{4}$$

$$H_c = \sum_{j=0}^{\infty} \sum_{\sigma} \left[ -t \, d^\dagger_{j+1,\sigma} d_{j,\sigma} + \Delta \, d_{j,\uparrow} d_{j,\downarrow} + \text{H.c.} \right], \tag{5}$$

$$H_{\text{imp}} = J \vec{S} \cdot \vec{s}_0, \tag{6}$$

where the operators $d_{j,\sigma}, d^\dagger_{j,\sigma}$ represent, respectively, annihilation and creation operators at the site $j = 0, 1, 2, \ldots$ of the chain with spin projection $\sigma$, and obey usual anti-commutation

relations $\left\{ d_{i,\sigma}, d_{j,\sigma'}^\dagger \right\} = \delta_{i,j}\delta_{\sigma,\sigma'}$. The parameters $t$ and $\Delta$ are, respectively, the effective hopping amplitude and the BCS pairing potential.

The theoretical model described above exhibits full SU(2)-spin rotation symmetry. The first step in our theoretical approach is to generalize the spin symmetry group from SU(2) to SU($N$), where the spin index $\sigma = \uparrow, \downarrow$ is replaced by the index $\alpha = 1, \ldots, N$. Here $N$ can take any arbitrary integer value. This allows to define the model in the $N \to +\infty$ limit, where a static mean-field theory becomes exact (i.e., saddle-point approximation, see Sec. 2.1) with $1/N$ being a small parameter that controls the magnitude of fluctuations [28, 30, 38].

In the normal metal case, the generalization from SU(2) to SU($N$) symmetry, apart from a rescaling of the Kondo exchange coupling, is rather straightforward [28–30]. The resulting large-$N$ approach provides a reasonably good description of the Kondo resonance and some of the low-energy properties of strong-coupling fixed point Hamiltonian of the Kondo model [28, 30, 39]. However, in the superconducting case a naïve generalization of the BCS pairing potential in $H_c$ (cf. Eq. (5)) to e.g. $H_\Delta = \Delta \sum_j \sum_{\alpha,\beta=1}^N \left[ d_{j\alpha} d_{j\beta} + \text{H.c.} \right]$ yields a SU($N > 2$) symmetry-breaking perturbation.[1] Physically, in an $N$-component Fermi gas the generalization of the (spin-singlet) Cooper pairs are $N$-particle bound states [40]. Therefore, the generalization of the BCS pairing potential is an $N$-fermion interaction of the form $H_\Delta^N = \Delta_N \sum_j \sum_{\alpha_1,\ldots,\alpha_N=1}^N \epsilon_{\alpha_1\cdots\alpha_N} d_{j\alpha_1} \cdots d_{j\alpha_N} + \text{H.c.}$, where $\epsilon_{\alpha_1\alpha_2\cdots\alpha_N}$ is the $N$-component fully anti-symmetric (Levi-Civita) symbol. For $N > 2$, $H_\Delta^N$ is clearly not quadratic and thus, in general, the resulting Hamiltonian does not describe an exactly solvable "mean-field" theory. A way out of this conundrum is to exploit the particle-hole symmetry of the impurity problem in the wide-band limit and map the BCS pairing Hamiltonian to a band insulator by means of the Bogoliubov transformation described in Appendix A for a general bipartite lattice. The bipartite lattice realizes the particle-hole symmetry on a lattice and thus the transformation turns the BCS pairing potential into a staggered lattice potential that leaves the form of the spin operators unchanged. For the one-dimensional model introduced above in Eq. (6), the transformation takes the form (see e.g. Ref. [21]):

$$c_{2j,\uparrow} = \frac{1}{\sqrt{2}}\left( d_{2j,\uparrow} + d_{2j,\downarrow}^\dagger \right), \tag{7}$$

$$c_{2j,\downarrow} = \frac{1}{\sqrt{2}}\left( d_{2j,\uparrow}^\dagger - d_{2j,\downarrow} \right), \tag{8}$$

$$c_{2j+1,\uparrow} = \frac{1}{\sqrt{2}}\left( d_{2j+1,\uparrow} - d_{2j+1,\downarrow}^\dagger \right), \tag{9}$$

$$c_{2j+1,\downarrow} = \frac{-1}{\sqrt{2}}\left( d_{2j+1,\uparrow}^\dagger + d_{2j+1,\downarrow} \right). \tag{10}$$

While this transformation preserves the form of the term $H_{\text{imp}}$ in Eq. (6), the transformed Hamiltonian of the host in terms of the $c$-operators becomes:

$$H_c = -t\sum_\sigma \sum_{j=0}^\infty \left( c_{j+1,\sigma}^\dagger c_{j,\sigma} + \text{H.c.} \right) + \sum_\sigma \sum_{j=0}^\infty (-1)^j \Delta c_{j,\sigma}^\dagger c_{j,\sigma}. \tag{11}$$

Note that the transformed $H_c$ lacks particle-hole symmetry in the $c$-operator basis (i.e. it is not invariant under $c_{j\sigma} \to (-1)^j c_{j,-\sigma}^\dagger$). This is not a problem for the generalization of the model to SU($N$) required below, and nevertheless the original particle-hole symmetry in the $d$-fermion basis can be recovered by undoing the transformation.

---

[1]Technically speaking, in SU($N > 2$) any fermion bilinear constructed from the $d$-operators is a rank-2 tensor which does not transform as a scalar and therefore breaks the required SU($N$) symmetry of the Hamiltonian, just like adding a magnetic field breaks the SU(2)-symmetry (i.e. spin-$\frac{1}{2}$) fermions.

Moreover, since in Eq. (11) the BCS pairing potential becomes a potential that couples to the total occupation of $c$-fermions at each lattice site, it is automatically a scalar under SU(2)-spin rotations and therefore admits a straightforward generalization to SU($N$) upon replacing the summation over $\sigma = \uparrow, \downarrow$ by one over $\alpha = 1, \ldots, N$. The full Hamiltonian generalized with SU($N$)-symmetry reads

$$H = \sum_{j=0}^{\infty} \sum_{\alpha=1}^{N} \left[ -t \left( c_{j+1,\alpha}^{\dagger} c_{j,\alpha} + \text{H.c.} \right) + \Delta (-1)^j c_{j,\alpha}^{\dagger} c_{j,\alpha} \right] - \frac{J}{N} \sum_{\alpha,\beta=1}^{N} : \left( f_{\alpha}^{\dagger} c_{0,\alpha} \right) \left( c_{0,\beta}^{\dagger} f_{\beta} \right) : , \quad (12)$$

where, in addition to rescaling $J \to J/N$, we have represented the SU($N$) generalization of the impurity spin operators in terms of pseudo-fermion $f$-operators (see Appendix B), which are subject to a constraint given in Eq. (B.3). In the above expression $: \ldots :$ stands for normal ordering of the product of fermion operators represented by the ellipsis $(\ldots)$. Writing the Kondo coupling in this form generates a scattering potential, which can be dropped because it induces a phase shift. The latter is of no physical consequence since it does not depend on the mean-field variational parameters to be introduced below [30]. From here on, we shall closely follow the derivation for the normal metal case [30] and use the path-integral formulation of this problem. Thus, we carry out a Hubbard–Stratonovich transformation of the interaction with the magnetic impurity:

$$\frac{J}{N} \sum_{\alpha,\beta=1}^{N} \left( \bar{f}_{\alpha} c_{0,\alpha} \right) \left( \bar{c}_{0,\beta} f_{\beta} \right) \to \sum_{\alpha=1}^{N} \left[ \bar{V} \left( \bar{f}_{\alpha} c_{0,\alpha} \right) + V \left( \bar{c}_{0,\alpha} f_{\alpha} \right) \right] + N \frac{\bar{V} V}{J} , \quad (13)$$

where $\bar{V}$ and $V$ are decoupling $U(1)$ bosonic fields which can be expressed as $V = |V| e^{i\phi}$. The partition function of the system written as follows:

$$Z = \int \mathcal{D} \left[ \bar{V}, V, \lambda \right] \int \mathcal{D} \left[ \bar{\psi}, \psi \right] e^{-\mathcal{S} \left[ \bar{\psi}, \psi, \bar{V}, V, \lambda \right]} , \quad (14)$$

where we have used the compact notation $\psi \equiv (\{c\}, \{f\})$ to represent the Grassmann variables inside the path-integral. The Euclidean action in the exponent of the integrand is defined as:

$$\mathcal{S} \left[ \bar{\psi}, \psi, \bar{V}, V, \lambda \right] = \sum_{\alpha=1}^{N} \int_0^{\beta} d\tau \, \bar{f}_{\alpha} \partial_{\tau} f_{\alpha} + \sum_{\alpha=1}^{N} \sum_{k,\nu=\pm} \int_0^{\beta} d\tau \, \bar{c}_{k,\nu,\alpha} \partial_{\tau} c_{k,\nu,\alpha} \int_0^{\beta} d\tau \, H \left[ \bar{\psi}, \psi, \bar{V}, V, \lambda \right], \quad (15)$$

with

$$H \left[ \bar{\psi}, \psi, \bar{V}, V, \lambda \right] = \sum_{\alpha=1}^{N} \left[ \sum_{k,\nu=\pm} \epsilon_{k,\nu} \bar{c}_{k,\nu,\alpha} c_{k,\nu,\alpha} + \lambda \bar{f}_{\alpha} f_{\alpha} + \left( \bar{V} \bar{f}_{\alpha} c_{0,\alpha} + V \bar{c}_{0,\alpha} f_{\alpha} \right) \right] + N \left( \frac{|V|^2}{J} - \lambda q \right). \quad (16)$$

In the above expressions we have introduced the eigenmodes of the clean insulator which obey the relation $[H_c, c_{k,\nu,\alpha}] = \epsilon_{k,\nu} c_{k,\nu,\alpha}$, with quantum number $\nu$ representing the valence ($\nu = -1$) or conduction ($\nu = +1$) band of the effective insulator. In addition, we have introduced the Lagrange multiplier $\lambda$ in order to inforce Eq. (B.3) by projecting the $f$-fermion occupation onto the physical sector $q = Q/N = 1/2$ [28, 30].

The interior integral in Eq. (14) defines the effective action $\mathcal{S}_{\text{eff}} \left[ \bar{V}, V, \lambda \right]$ through the relation:

$$e^{-\mathcal{S}_{\text{eff}} \left[ \bar{V}, V, \lambda \right]} = \int \mathcal{D} \left[ \bar{\psi}, \psi \right] e^{-\mathcal{S} \left[ \bar{\psi}, \psi, \bar{V}, V, \lambda \right]} . \quad (17)$$

Since the action $\mathcal{S}\left[\bar{\psi}, \psi, \bar{V}, V, \lambda\right]$ is extensive in $N$, in the limit $N \to \infty$ the effective action $\mathcal{S}_{\text{eff}}$ is dominated by the saddle point, which can be found by extremizing the effective action:

$$\frac{\delta \mathcal{S}_{\text{eff}}}{\delta \bar{V}(\tau)} = \frac{1}{N} \sum_{\alpha=1}^{N} \langle \bar{f}_{\alpha}(\tau) c_{0,\alpha}(\tau) \rangle + \frac{V(\tau)}{J} = 0, \tag{18}$$

$$\frac{\delta \mathcal{S}_{\text{eff}}}{\delta \lambda(\tau)} = \frac{1}{N} \sum_{\alpha=1}^{N} \langle \bar{f}_{\alpha} f_{\alpha} \rangle - q = 0. \tag{19}$$

In the next section we analyze in detail these saddle-point equations.

## 2.1 Analysis of the large-$N$ extrema

We shall solve the problem in the radial gauge where

$$f_{\alpha}(\tau) \to e^{i\phi(\tau)} f_{\alpha}(\tau), \quad V(\tau) \to e^{i\phi(\tau)} V(\tau), \quad \lambda(\tau) \to \lambda - i\partial_{\tau}\phi(\tau),$$

absorbing the U(1) phase fluctuations of $V$ in a (now dynamical) variable $\lambda(\tau)$ [30]. Next, we focus on the static limit of all the bosonic fields, and the Hamiltonian $H\left[\bar{\psi}, \psi, \bar{V}, V, \lambda\right]$ in Eq. (16) reduces to $H\left[\bar{\psi}, \psi, \bar{V}, V, \lambda\right] \to H_{\text{MF}}$, where $H_{\text{MF}}$ is a straightforward mean-field Hamiltonian:

$$H_{\text{MF}} = \sum_{\alpha=1}^{N} \left[ \sum_{k, \nu=\pm} \epsilon_{k,\nu} \bar{c}_{k,\nu,\alpha} c_{k,\nu,\alpha} + \lambda \bar{f}_{\alpha} f_{\alpha} + V\left(\bar{f}_{\alpha} c_{0,\alpha} + \bar{c}_{0,\alpha} f_{\alpha}\right) \right] + N\left(\frac{|V|^2}{J} - \lambda q\right), \tag{20}$$

which allows to regard the system as an insulator with a resonant level $f_{\alpha}$ with effective on-site energy $\lambda$, and coupled to the insulating host via a hybridization parameter $V$. The free-energy of this effective model, defined from $Z_{\text{MF}} = e^{-\beta F_{\text{MF}}}$ [see Eq. (17)], can be computed [30] and reads:

$$F_{\text{MF}} = -\frac{1}{\beta} \sum_{i\nu_n} \text{Tr} \ln\left[-\mathcal{G}^{-1}(i\nu_n)\right] + N\left(\frac{|V|^2}{J} - \lambda q\right). \tag{21}$$

In this expression we have defined the fermionic Matsubara Green's function matrix $\mathcal{G}^{-1}(i\nu_n) = [i\nu_n - H_{\text{MF}}]$, with $\nu_n = \pi(2n+1)/\beta$. Since we are only interested in the local physics at the impurity site, we subtract the contribution of $H_c$

$$\Delta F_{\text{MF}} = F_{\text{MF}} - F_c^{(0)} = -\frac{1}{\beta} \sum_{i\nu_n} \text{Tr} \ln\left[-\mathcal{G}_{ff}^{-1}(i\nu_n)\right] + N\left(\frac{|V|^2}{J} - \lambda q\right), \tag{22}$$

where $\mathcal{G}_{ff}(i\nu_n)$ is the $f$-electron Matsubara Green's function [41]

$$\begin{aligned}
\mathcal{G}_{ff}(i\nu_n) &= -\int_0^\beta d\tau\, e^{i\nu_n\tau} \langle T_\tau f_\alpha(\tau) f_\alpha^\dagger(0) \rangle \\
&= \frac{1}{i\nu_n - \lambda_0 - V^2 \mathcal{G}_{cc}^{(0)}(i\nu_n)}.
\end{aligned} \tag{23}$$

Here $T_\tau$ is the imaginary-time ordering operator, and $\mathcal{G}_{cc}^{(0)}(z)$ is the Green's function of the clean insulator at site $j=0$, whose expression for the semi-infinite one-dimensional tight-binding chain can be analytically obtained, e.g. using the recursion method [42]:

$$\mathcal{G}_{cc}^{(0)}(z) = \frac{z+\Delta}{2t^2} \pm \sqrt{\left(\frac{z+\Delta}{2t^2}\right)^2 - \frac{1}{t^2}\frac{z+\Delta}{z-\Delta}}, \tag{24}$$

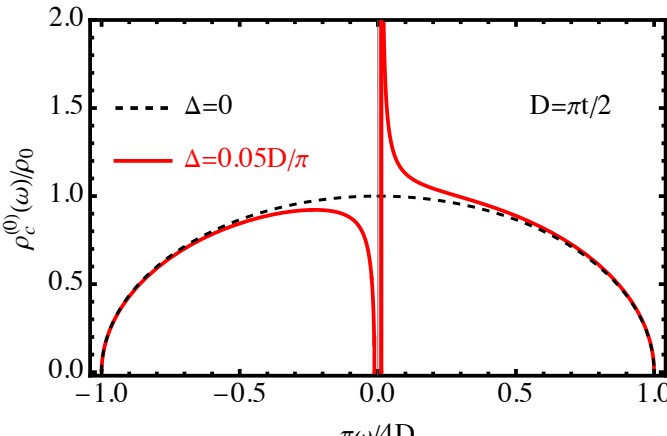

Figure 1: (Color online). Local density of states at site $j = 0$ in the tight-binding chain without impurity for the normal metal case $\Delta = 0$ (black dashed line), and for the effective insulator case (red line) with $\Delta = 0.05D/\pi$. Here $\rho_0 = 1/2D$ and $\rho_c^{(0)}(\omega) = -\frac{1}{\pi}\text{Im}\left[\mathcal{G}_{cc}^{(0)}(\omega^+)\right]$, where $\omega^+ = \omega + i0^+$, $0^+$ being a positive infinitesimal.

where the sign must be chosen such that for $\text{Im}[z] > 0$, $\text{Im}\left[\mathcal{G}_{cc}^{(0)}(z)\right] < 0$. In the above expression, employing the the SU($N$) symmetry, we have dropped the index $\alpha$.

In order to compare the large-$N$ and the NRG approaches, we assume that the properties of the impurity in the (superconducting) host are scaling functions of a single parameter, namely the ratio of gap $\Delta$ to the Kondo temperature, $T_K$ (this scaling assumption has been verified both experimentally and theoretically, see e.g. Refs. [43, 44] and references therein). The former is an energy scale that characterizes the host in the absence of magnetic impurities, whilst the later characterizes the impurity in the normal state of the host (i.e. for $\Delta = 0$, see discussion in Sec. 3.1 for more details about its definition). Since for $T_K \ll D$ the Kondo temperature depends on the the dimensionless coupling $\rho_0 J$, where $\rho_0$ is density of states of the host at the Fermi energy for $\Delta = 0$ and $J$ the exchange coupling, we choose $\rho_0$ to the same in both approaches. Thus, recalling that the constant density of states used in NRG is $\rho_0 = 1/2D$ (with $D$ the band width used in the NRG calculations), we require:

$$-\frac{1}{\pi}\text{Im}\left[\mathcal{G}_{cc}^{(0)}(z \to \omega^+, \Delta = 0)\right]_{\omega=0} = \frac{1}{\pi t} = \frac{1}{2D}, \tag{25}$$

where $\omega^+ = \omega + i0^+$ ($0^+$ denoting a positive infinitesimal).

For illustration purposes, in Fig. 1 we show the unperturbed local density of states (LDOS) at site $j = 0$ in the chain, $\rho_c^{(0)}(\omega) = -\frac{1}{\pi}\text{Im}\left[\mathcal{G}_{cc}^{(0)}(\omega^+)\right]$, both in the normal case $\Delta = 0$ (black dashed line) and superconducting case $\Delta > 0$ (continuous red line). In this latter case, we can see the presence of a gap $2\Delta$ in the single-particle excitation spectrum. We note the asymmetry of the plot in the case $\Delta > 0$, due to the breaking of the particle-hole symmetry by the staggered potential in Eq. (11). As mentioned in the preceding section, the particle-hole symmetry of the original model can be restored undoing the transformation in Eqs. (7)-(10), and expressing the LDOS in terms of the original $d-$fermions.

Using Eq. (22), the extrema equations (18) and (19) become,

$$\frac{\partial \Delta F_{\text{MF}}}{\partial V} = V\left[\frac{1}{J} + \frac{1}{\beta}\sum_{i\nu_n}\mathcal{G}_{ff}(i\nu_n)\mathcal{G}_{cc}^{(0)}(i\nu_n)\right] = 0, \tag{26}$$

$$\frac{\partial \Delta F_{\text{MF}}}{\partial \lambda} = -q + \frac{1}{\beta}\sum_{i\nu_n}\mathcal{G}_{ff}(i\nu_n) = 0. \tag{27}$$

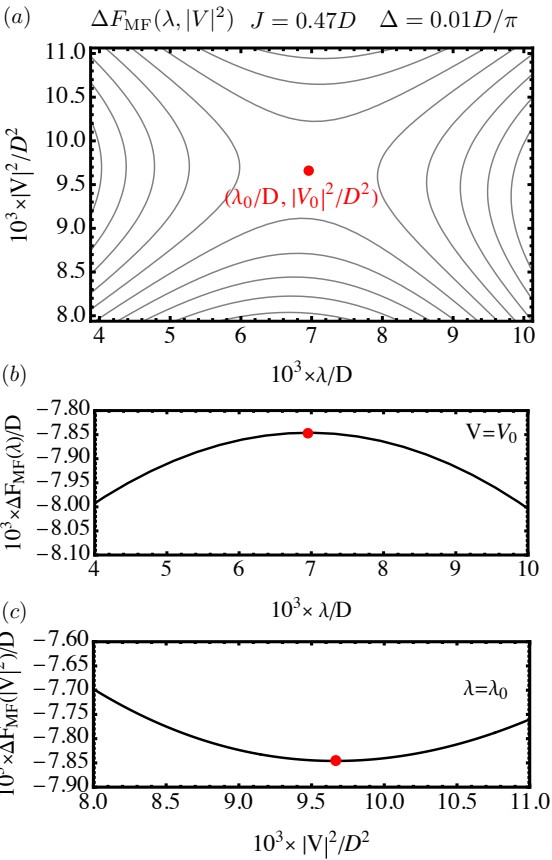

Figure 2: (Color online). Impurity free energy near a physical (saddle-point) solution of Eqs. (26) and (27) for $J = 0.47D, \Delta = 0.01D/\pi$. The red points are the location of the solutions $V_0, \lambda_0$. (a) Contour plot of the impurity free energy. The physical solution is a saddle point of the free energy. Panels (b) and (c) are the free-energy as a function of $\lambda$ and $|V|^2$ with fixing $|V|^2 = |V_0|^2$ and $\lambda = \lambda_0$, respectively.

Note that $V = 0$ and $\lambda = 0$ always correspond to extrema, which describes a decoupled $f$-level from the host, or in the language of the original model, an unscreened impurity [1, 2]. At $T = 0$, the Matsubara sums above can be evaluated by contour integration on the complex plane. Thus, we obtain the following expressions (assuming $V \neq 0$):

$$\frac{1}{J} - \frac{1}{\pi} \int_{-\infty}^{0} d\omega \, \text{Im}\left[\mathcal{G}_{cc}^{(0)}\left(\omega^+\right)\mathcal{G}_{ff}\left(\omega^+\right)\right] = 0\,, \tag{28}$$

$$-\frac{1}{\pi} \int_{-\infty}^{0} d\omega \, \text{Im}\left[\mathcal{G}_{ff}\left(\omega^+\right)\right] - q = 0\,. \tag{29}$$

For every pair of microscopic parameters $J, \Delta$ in the Hamiltonian of Eq. (4), the above expressions define a system of nonlinear coupled equations which yield the extrema of the large-$N$ effective action where $V = V_0$, $\lambda = \lambda_0$. In practice, we have solved these equations using a numerical implementation of the Newton-Raphson algorithm. Thus, although we are interested in the $T = 0$ case, for reasons of numerical stability we have used a small but finite absolute temperature $T = \Delta/200$. This smoothens the non-analyticity introduced by the YSR state near $\omega = 0$ at $T = 0$ due to the sharpness of the Fermi-Dirac occupation of the $f$-level, which results in numerical instabilities (see Appendix C for details).

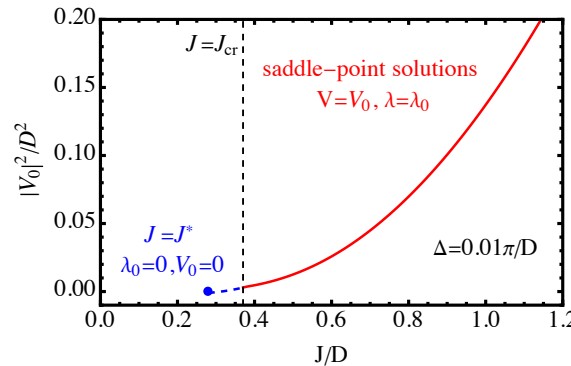

Figure 3: (Color online) Plot of $V_0$ as a function of the exchange parameter $J$, derived by solving the conditions for extrema of the large-$N$ effective action (28) and (29). The blue dot corresponds to $J^*$ (see Eq. (31)). The blue dashed curve are extrema that correspond to local maxima of the large-$N$ effective action. The continuous red curve corresponds to true saddle point solutions. $J_{cr}$ is the minimum value of the Kondo coupling for which the saddle point solutions exist.

In contrast to the normal metal case, a distinct aspect of this problem is the existence of a gap in the excitation spectrum of the host. This feature drastically changes the low-energy properties of the system, and allows for the existence of a finite value of the exchange coupling $J = J^* > 0$ for which Eqs. (28) and (29) are solved by $V_0 = 0$ and $\lambda_0 = 0$. Imposing the condition $V = 0$ in Eq. (29), and recalling that $q = 1/2$, we find:

$$\frac{1}{2} = \int_{-\infty}^{0} d\omega\, \delta(\omega - \lambda_0), \tag{30}$$

whose only possible solution is $\lambda_0 = 0$. Hence, $J^*$ is obtained by setting $V = V_0 = 0$ and $\lambda = \lambda_0 = 0$ in Eq. (29), which yields:

$$\frac{1}{J^*} = \frac{1}{\pi} \int_{-\infty}^{0} d\omega\, \mathrm{Im}\left[ \frac{\mathcal{G}_{cc}^{(0)}(\omega^+)}{\omega^+} \right]. \tag{31}$$

For $J > J^*$, solutions with $V_0 \neq 0$ and $\lambda_0 \neq 0$ can be found to the above extrema conditions, Eqs. (28) and (29). However, for $J^* < J < J_{cr}$, the extrema correspond to local maxima of the free energy. Here $J_{cr}$ is defined as the minimum value of the Kondo exchange for which $V_0$ and $\lambda_0$ are true saddle-points of the free energy (as e.g. shown Fig. 2). The latter is a defining feature of a physical ground-state solution, which in the present case corresponds to the Kondo screened phase. The situation is summarized in Fig. 3, which illustrates that $J^*$ is connected to $J_{cr} > J^*$ by a string of local maxima (dashed blue curve) and the extrema conditions only settle onto true saddle-pointd with $V = V_0 \neq 0$ (and $\lambda = \lambda_0 \neq 0$, not shown) for $J > J_{cr}$ (continuous red curve). Thus, regarding $V_0, \lambda_0$ the transition is discontinuous, which agrees with the fact that for the original SU(2)-symmetric system it is a level crossing transition [45]. In hindsight, the evolution of the extrema from local maxima to saddle-points can be seen as the consequence of the necessity of the system to undergo a discontinuous phase transition between the unscreened phase ($V_0 = 0$) and the Kondo screened phase ($V_0 \neq 0$) with the parameter $V_0$ as a continuous function of $J$.

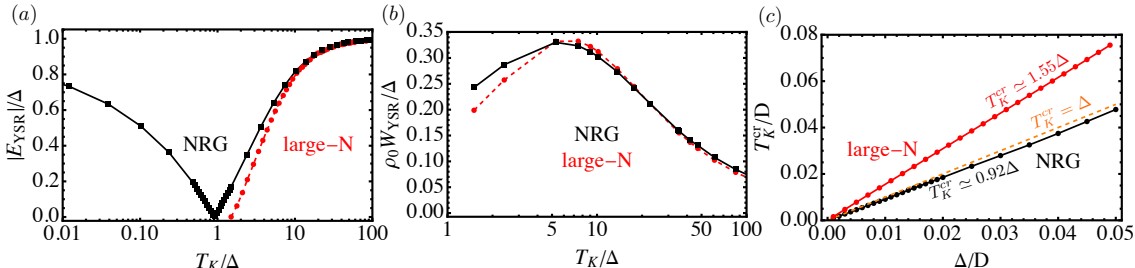

Figure 4: (Color online) (a) Position of the YSR peaks (absolute value) as a function of the Kondo temperature obtained from the NRG method (black dots) and the large-$N$ mean-field theory (red dots) as a function of $T_K/\Delta$, computed for parameters $\Delta = 0.001D$ for NRG and $\Delta = 0.01D/\pi$ for large-$N$ calculations. The point where the YSR level crosses the Fermi energy $E_F = 0$ corresponds to the critical point indicating the doublet-singlet transition. (b) Spectral weights of YSR peaks, $W_{\text{YSR}} = W_\uparrow(E_{\text{YSR}}) + W_\downarrow(E_{\text{YSR}})$(c.f. Eq. (D.5)), at strong coupling obtained from NRG (black dots) and large-$N$ theory (red dots). (c) Comparison of the critical points $T_K^{\text{cr}}/\Delta$ vs $\Delta$ from both methods. The linear behavior in both methods expresses the universality and robustness of the transition. While NRG yields $T_K^{\text{cr}} \simeq 0.92\Delta$, which is consistent with previous works [37, 46], the saddle-point approximation overestimates this dependence.

## 3 Results

### 3.1 Intra-gap YSR excitations

After solving the equations for the extrema and obtaining the set of physical saddle-point solutions $V_0, \lambda_0$, we next focus on the calculation of observable properties. One such quantity is the local density of states (LDOS), which can be measured from the differential conductance signal of a STM. Returning for a while to the SU(2)-symmetric system, we recall the definition of the LDOS:

$$\rho_d(\omega) = -\frac{1}{\pi}\text{Im}\left[\sum_\sigma \mathcal{G}_{dd,\sigma}(\omega^+)\right], \tag{32}$$

where the propagator $\mathcal{G}_{dd,\sigma}(z)$ corresponds to the propagator of the original $d$-fermions. Using the Bogoliubov transformation Eqs. (7) to (10), we obtain the relation

$$\sum_\sigma \mathcal{G}_{dd,\sigma}(z) = \frac{1}{2}\sum_\sigma\left[\mathcal{G}_{cc,\sigma}(z) + \bar{\mathcal{G}}_{cc,\sigma}(z)\right], \tag{33}$$

where $\bar{\mathcal{G}}_{cc,\sigma}(z)$ is the hole propagator computed from the analytical continuation to complex frequency $z$ of

$$\bar{\mathcal{G}}_{cc,\sigma}(i\nu_n) = \int_0^\beta d\tau e^{i\nu_n\tau}\langle T_\tau c_{0,\sigma}^\dagger(\tau)c_{0,\sigma}(0)\rangle. \tag{34}$$

By SU(2) symmetry, the Green's functions at both sides are independent of spin, which means the spin index can be dropped. Thus, we arrive at the relation:

$$\begin{aligned}\mathcal{G}_{dd}(z) &= \frac{1}{2}\left[\mathcal{G}_{cc}(z) + \bar{\mathcal{G}}_{cc}(z)\right]\\ &= \frac{1}{2}\left[\mathcal{G}_{cc}(z) - \mathcal{G}_{cc}(-z)\right],\end{aligned} \tag{35}$$

where we have used the property $\bar{\mathcal{G}}_{aa}(z) = -\mathcal{G}_{aa}(-z)$. Furthermore, from the equations of motion of the $c$-fermions, the exact electron propagator can be expressed as [47]:

$$\mathcal{G}_{cc}(z) = \mathcal{G}_{cc}^{(0)}(z) + \mathcal{G}_{cc}^{(0)}(z)\mathcal{T}(z)\mathcal{G}_{cc}^{(0)}(z), \tag{36}$$

where $\mathcal{T}(z)$ is the $T$-matrix, which can be obtained from the following expression:

$$\mathcal{T}_\sigma(i\nu_n) = -\int_0^\beta d\tau e^{i\nu_n\tau}\langle T_\tau \mathcal{O}_\sigma(\tau)\mathcal{O}_\sigma^\dagger(0)\rangle, \tag{37}$$

where $\mathcal{O}_\sigma = \frac{J}{2}\left[c_{0,-\sigma}S^{-\sigma} + \sigma c_{0,\sigma}S^z\right]$.

Up to this point, the above derivation is formally exact. Let us turn to the equation for the $c$-fermion Green's function within the saddle-point approximation, which can be also obtained by the equations-of-motion method and reads:

$$\mathcal{G}_{cc}(z) = \mathcal{G}_{cc}^{(0)}(z) + \mathcal{G}_{cc}^{(0)}(z)\left[V_0^2 \mathcal{G}_{ff}(z)\right]\mathcal{G}_{cc}^{(0)}(z). \tag{38}$$

Comparing the exact expression Eq. (36), and Eq. (38), we see that within the saddle-point approximation, i.e. to leading order in $1/N$, $\mathcal{T}_\sigma(z) \simeq V_0^2 \mathcal{G}_{ff}(z)$. This relation is also expected from the pseudo-fermion spin representation of the impurity spin in Eq.(12). Indeed, replacing the operator $Jc_{0,-\sigma}S^{-\sigma}$ in the above definition of $\mathcal{O}_\sigma$ by $Jc_{0,-\sigma}S^{-\sigma} \to Jc_{0,\alpha}\left(f_\alpha^\dagger f_\beta\right) \simeq J\langle c_{0,\alpha}f_\alpha^\dagger\rangle f_\beta = -V_0 f_\beta$ where the saddle-point Eq. (18) has been used, and therefore we obtain the same result, namely $\mathcal{T}_\sigma(z) \simeq V_0^2 \mathcal{G}_{ff}(z)$ as above. Comparing to the NRG results for the SU(2)-symmetric system we can assess the magnitude of the ($1/N$, etc.) fluctuation corrections to the $N \to +\infty$ saddle-point approximation.

Finally, using the Eqs. (35) and (38), the change in the LDOS due to the impurity after subtracting the background (i.e., the contribution of the bare $c-$fermion propagator) can be obtained and yields the following expression:

$$\Delta\rho_d(\omega) = \Delta\rho_c(\omega) + \Delta\rho_c(-\omega), \tag{39}$$

where:

$$\Delta\rho_c(\omega) = -\frac{1}{\pi}\mathrm{Im}\left[V_0^2\left[\mathcal{G}_{cc}^{(0)}(\omega^+)\right]^2 \mathcal{G}_{ff}(\omega^+)\right]. \tag{40}$$

Note that particle-hole symmetry of the original model, Eq. (6), has been restored in Eq.(39).

In addition, from Eq. (40), and since the $c$-fermion Green's function $\mathcal{G}_{cc}^{(0)}(z)$ has no singularities inside the gap, the intra-gap YSR states must emerge from the poles of the retarded Green's function $\mathcal{G}_{ff}(\omega^+)$ in the region $-\Delta < \omega < \Delta$. Therefore, from Eq. (23), we obtain the equation for the energy of the ingap YSR states

$$E_{\mathrm{YSR}} - \lambda_0 - V_0^2 \mathrm{Re}\left[\mathcal{G}_{cc}^{(0)}(E_{\mathrm{YSR}})\right] = 0. \tag{41}$$

In Fig. 4(a), we show the position of the YSR states derived from Eq. (41), and from the NRG method. In the case of the NRG, the position of $E_{\mathrm{YSR}}$ is extracted from the position of the YSR peaks in the spectral functions (see e.g. Fig. 5, and Appendix E for more details about the calculation of the spectral functions within the NRG method). In order to compare our results with the experiment and with other theoretical approaches, we plot the position of the YSR states as a function of the dimensionless ratio $T_K/\Delta$. To this end, we define the Kondo temperature (in $k_B = 1$ units) $T_K$ as the half-width at half-maximum (HWHM) of the spectral functions in the normal state following Ref. [47]. In this regard, it is important to recall that the expression for the Kondo temperature $T_K = De^{-1/J\rho_0}$, which is frequently used in the literature, is only valid in the weak-coupling regime where $\rho_0 J \ll 1$ [48], and it is not valid in

the strong coupling regime of interest to us here. Plotting physical quantities in terms of the ratio $T_K/\Delta$ is relevant to experiments, in which $T_K$ can be directly extracted from the width of the Kondo resonance in the STM differential conductance. This choice also allows us to compare results from different theoretical approaches, for which the details of the density of states and other observables, may be different, but the ratio $T_K/\Delta$ is the same. Indeed, this is the case for the comparison between the NRG and large-$N$ results reported below.

Within the large-$N$ approach, we have computed the Kondo temperature $T_K$ by first solving the saddle-point equations, (28) and (29) for the normal system (i.e. for $\Delta = 0$), and then extracted the HWHM from $\rho_f(\omega) = -\mathrm{Im}\,\mathcal{G}_{ff}(\omega^+)/\pi$, where Eq. (23) is used (this last step is required as the line-shape of the Kondo peak is not Lorentzian because the density of states for $\Delta = 0$ is not constant, see dashed curve in Fig. 1).

As discussed above, our large-$N$ approach is able to describe the phase transition from the unscreened to the Kondo screened phase (see Figs. 4 (a,c)). Intuitively speaking, this transition occurs when the coupling to the impurity (corresponding to the energy scale $T_K$) is of the same order as the pairing gap $\Delta$, and therefore the exchange interaction is able to break the Cooper pairs, allowing the impurity to bind (an odd number of) quasi-particles that collectively screen the impurity spin. Note that the saddle-point approximation overestimates the transition point and yields $T_K^{\mathrm{cr}}/\Delta \simeq 1.55$, which is higher than the NRG result of $T_K^{\mathrm{cr}}/\Delta \simeq 0.92$. In the NRG the transition occurs at the point where singlet and doublet ground states cross, while, within the large-$N$ approach, it occurs where the first physical (i.e. true saddle-point of the free energy) solution appears. Since the large-$N$ approach is an intrinsically variational method, we believe that this overestimation is rooted at the overestimation of the singlet ground-state energy, which is directly related to the energy of the YSR in the screened phase. By the variational principle, the minimization with respect to $V$ under the constraint imposed by $\lambda$ yields a saddle-point free energy which must be larger than, or equal to, the actual ground-state energy. Since the transition corresponds to a singlet-doublet level crossing, near the transition point quantum fluctuations contribute to lower the actual ground state energy. In the saddle-point approximation, however, such fluctuations are neglected and lead to an overestimation of the singlet ground state energy. On the other hand, the positions of the YSR peaks from the large-$N$ theory converge to the NRG result at $T_K \gtrsim 10\Delta$ since in that case fluctuations are suppressed by the strong Kondo coupling. The same tendency is observed in the spectral weight shown in Fig. 4(b), where the discrepancies between the two approaches become is smaller for $T_K \gtrsim 10\Delta$.

Another feature shown in Fig. 4 is the lack of physical solutions within the large-$N$ saddle-point approach in the region $T_K < T_K^{\mathrm{cr}}$. This is a reflection of the failure to describe the unscreened phase within this approach. In fact, in this regime the only solution to Eqs. (26) and (27) is $V_0 = \lambda_0 = 0$, which describes an $f$-level decoupled from the host. For $N = 2$ and $V_0 = 0$, the mean-field Hamiltonian, Eq. (20) has a doubly degenerate ground state corresponding to the two possible spin orientations of the $f$-fermion. This state is adiabatically connected with the $J = 0$ ground state of the original system. However, the spectrum of mean-field Hamiltonian does not contain ingap states. We speculate that this feature will be cured if fluctuations were taken into account. While this is an evident drawback of the present approach, we note that it performs increasingly well as $T_K/\Delta$ becomes large into the strong coupling regime, thus providing a reliable analytical tool which can complement other (e.g. perturbative) approaches.

Finally, we stress that, while Figs. 4(a) and (b) have been obtained for the particular choices of $\Delta/D = 0.001$ in the case of NRG ($\Delta/D = 0.01/\pi$ for the large-$N$ approach), our results are robust and do not depend on the specific values of parameters. To show this and to benchmark the large-$N$ method, in Fig. 4(c) we show the transition point $T_K^{\mathrm{cr}}$ as a function of $\Delta$. The linear dependence is an indication of the robustness of the method, and the different slope of the

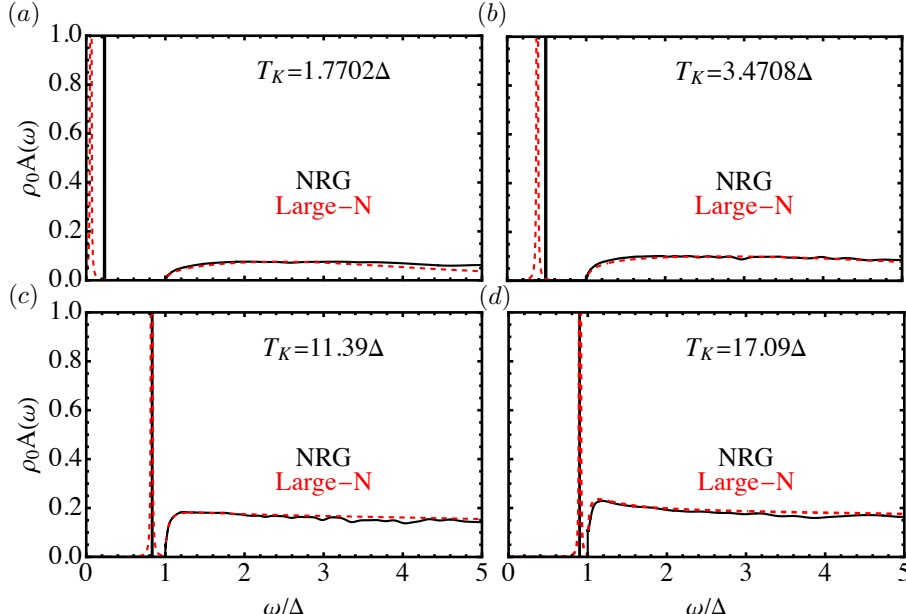

Figure 5: (Color online) Spectral function $A(\omega) = A_\uparrow(\omega) + A_\downarrow(\omega)$ obtained from NRG (black solid curves) and the large-$N$ theory (red dashed curves) for (a) $T_K = 1.7702\Delta$, (b) $T_K = 3.4708\Delta$, (c) $T_K = 11.39\Delta$ and (d) $T_K = 17.09\Delta$. Here $\rho_0 = 1/2D$. The finite width of YSR peaks for the large-$N$ approach is due to a finite broadening parameter $\delta = 10^{-6} D$. Here $\Delta = 10^{-3} D$ for NRG and $\Delta = 10^{-2} D/\pi$ for the large-$N$ calculations.

two lines reflects the overestimation of the singlet ground state energy within the saddle-point approximation, which has been discussed above.

## 3.2 Spectral function at the impurity site

Besides the position of the transition and energy of the YSR states, another experimentally relevant quantity that can be obtained using the large-$N$ approach is the spectral function at the impurity site, which is formally related to the $T$-matrix by means of the expression:

$$A_\sigma(\omega) = -\frac{1}{\pi} \text{Im}[\mathcal{T}_\sigma(\omega^+)], \tag{42}$$

where $\mathcal{T}_\sigma(z)$ is the $T$-matrix given in Eq. (37).

In Fig. 5 we show the spectral functions derived from both the NRG method and the large-$N$ approach. Both approaches are remarkable good agreement, particularly in the region where $T_K/\Delta \gtrsim 10$. The position of the YSR peaks follow the curves shown in Fig. 4(a), showing a discrepancy for $T_K/\Delta$ near the critical point, but the agreement becomes quantitatively accurate in the strong-coupling regime. This is also illustrated by the result of spectral weights in Fig. 4(b).

## 4 Conclusions

We have studied a spin-$\frac{1}{2}$ quantum impurity coupled to a conventional superconductor using a large-$N$ approach in the saddle-point approximation. This is a problem of both fundamental and practical interest, which is under active research in condensed-matter physics for its

implications for engineering and controlling exotic quantum states of matter and their excitations [49–51].

In normal metals, the large-$N$ method in the saddle-point approximation is a well-established and reliable approach for the description of Kondo impurities at low temperatures [30, 33]. Here we have shown how to extend this approach to superconductors by generalizing the SU(2)-spin symmetry of the Hamiltonian to SU($N$). The first step in this generalization requires mapping the problem to a magnetic impurity in an insulating host. This circumvents the problem of the pairing potential breaking the SU($N$) symmetry of a Hamiltonian. The resulting model has been analyzed in the large-$N$ limit using the saddle-point approximation. We have shown that this approach is capable of describing the transition to the Kondo screened phase, but unlike the normal metal case, the transition happens for a finite value of the Kondo coupling $J$, or more precisely, a finite value of the ratio of the Kondo temperature $T_K$ to the superconducting gap $\Delta$. In the strong coupling regime (i.e. $T_K/\Delta \gtrsim 1$), we have computed spectral properties such as the position of the YSR ingap states, their spectral weight, as well as the spectral function of the continuum states. However, near the transition point, $T_K \sim \Delta$, the saddle-point approximation overestimates the Kondo-singlet ground state energy due to the neglect of finite $N$ corrections.

In the weak coupling region $T_K < T_K^{\mathrm{cr}}$ the saddle-point approximation is not accurate, as the magnetic impurity effectively decouples from the superconductor. This is described by solutions of the free-energy extrema equations that correspond to a doublet ground state (for $N = 2$) emerging from an impurity level that is not hybridized with its host. However, this mean-field Hamiltonian is unable to describe the ingap YSR states as well as other spectral properties in the weak coupling regime. This is obviously a drawback of the method, which may be traced back to the static nature of the saddle-point approximation, that neglects quantum fluctuations. We speculate that accounting for fluctuation effects, which appear at higher orders in $1/N$, should provide a more accurate description of the unscreened phase in the weak coupling regime.

Nevertheless, despite the failure to accurately describe the weak coupling regime, in the strong-coupling regime where $T_K \gtrsim 10\Delta$, the large $N$ approach yields results that show a remarkable agreement with NRG for the energy of YSR states as well as in the spectral function at the impurity site. We believe this is due to the suppression of quantum fluctuations caused by the Kondo screening of impurity as the system moves into the strong coupling regime. This suppression helps to stabilize the Kondo singlet as the ground state of the system, away from the competition with BCS pairing correlations that takes place for $T_K \simeq \Delta$.

One major limitation of the present large-$N$ approach is the requirement that the models to be studied must exhibit particle-hole symmetry. Accounting for particle-hole symmetry breaking perturbations is desirable in order to provide a quantitative description of experimental systems. However, we must regard the present approach as a computationally affordable method (akin to the classical approach of YSR [3, 5, 52]) to obtain valuable (semi-)analytical insights into the real-space and spectral properties of strongly coupled magnetic impurities in systems for which other, sophisticated numerical tools such as NRG, DMRG, or quantum Monte Carlo may not be easily applicable. This is indeed the case of multiple impurity systems, impurity lattices of various dimensionalities (especially chains [34, 35]), superconductor-normal heterostructures [25, 36] as well as impurities in superconducting hosts with complex (but particle-hole) symmetric band structures. Such systems certainly provide an exciting playground for further exploration of the complex phenomena related to quantum magnetic impurities in superconductors.

## Acknowledgments

CHH acknowledges a PhD Fellowship granted by DIPC and AML acknowledges a temporary visitor appointment at DIPC, which kickstarted this collaboration.

**Funding information** This work has been supported supported by the Agencia Estatal de Investigación (AEI) of the Ministerio de Ciencia, Innovación, y Universidades (Spain) through AEI/10.13039/501100011033 Grants No. PID2020- 120614GB-I00 (ENACT) and No. PID2023-148225NB-C32 (SUNRISE).

## A  From BCS to band insulator on a bipartite lattice

In this Appendix we discuss the Bogoliubov transformation allowing for the SU($N$)-symmetric extension of the impurity model in a more general framework. The purpose is to illustrate how the BCS Hamiltonian can be mapped to an insulator model in a more general class of tight-biding Hamiltonians on bipartite lattices than the one dimensional chain described by $H_c$ in Eq. (6). In particular, we emphasize that, as long as the full system (i.e. host + impurity) exhibits particle-hole symmetry, the transformation can be used for systems of arbitrary dimensionality.

Let us consider the following BCS pairing Hamiltonian on a bipartite lattice, that is, a lattice consisting of two interpenetrating lattices $A$ and $B$:

$$H_c = H_0 + H_\Delta \,, \tag{A.1}$$

$$H_0 = -t \sum_{\langle \vec{R}, \vec{R'} \rangle, \sigma} \left[ d^\dagger_{A\vec{R}\sigma} d_{B\vec{R'}\sigma} + \text{H.c.} \right] \,, \tag{A.2}$$

$$H_\Delta = \Delta \sum_{\vec{R}} \sum_{p=A,B} \left[ d_{p\vec{R}\uparrow} d_{p\vec{R}\downarrow} + \text{H.c.} \right] \,. \tag{A.3}$$

This (mean-field) BCS pairing Hamiltonian can be mapped to one describing a band insulator where a gap $\propto \Delta$ is opened by a staggered lattice potential. This is achieved by means of the following Bogoliubov transformation:

$$c_{A\vec{R}\uparrow} = \frac{1}{\sqrt{2}} \left( d_{A\vec{R}\uparrow} + d^\dagger_{A\vec{R}\downarrow} \right) \,, \tag{A.4}$$

$$c_{A\vec{R}\downarrow} = \frac{1}{\sqrt{2}} \left( d^\dagger_{A\vec{R}\uparrow} - d_{A\vec{R}\downarrow} \right) \,, \tag{A.5}$$

$$c_{B\vec{R}\uparrow} = \frac{1}{\sqrt{2}} \left( d_{B\vec{R}\uparrow} - d^\dagger_{B\vec{R}\downarrow} \right) \,, \tag{A.6}$$

$$c_{B\vec{R}\downarrow} = \frac{-1}{\sqrt{2}} \left( d^\dagger_{B\vec{R}\uparrow} + d_{B\vec{R}\downarrow} \right) \,. \tag{A.7}$$

For the 1D chain with nearest neighbor hopping that was considered in Sec. 2, the $A$ sublattice corresponds to the even sites where $A\vec{R} \to 2j$, $j$ being an integer and the $B$ sublattice to the odd sites where $B\vec{R} \to 2j+1$. The inverse of the transformation reads:

$$d_{A\vec{R}\uparrow} = \frac{1}{\sqrt{2}} \left( c_{A\vec{R}\uparrow} + c^\dagger_{A\vec{R}\downarrow} \right) \,, \tag{A.8}$$

$$d_{A\vec{R}\downarrow} = \frac{1}{\sqrt{2}} \left( c^\dagger_{A\vec{R}\uparrow} - c_{A\vec{R}\downarrow} \right) \,, \tag{A.9}$$

$$d_{B\vec{R}\uparrow} = \frac{1}{\sqrt{2}}\left(c_{B\vec{R}\uparrow} - c^\dagger_{B\vec{R}\downarrow}\right),$$ (A.10)

$$d_{B\vec{R}\downarrow} = \frac{-1}{\sqrt{2}}\left(c^\dagger_{B\vec{R}\uparrow} + c_{B\vec{R}\downarrow}\right).$$ (A.11)

Let us first consider the transformation of the hopping term in Eq. (A.3). To see that it is left invariant, we consider the sum of the following two contributions:

$$-t\sum_{\langle\vec{R},\vec{R}'\rangle}\left[d^\dagger_{A\vec{R}\uparrow}d_{B\vec{R}'\uparrow} + d^\dagger_{B\vec{R}'\downarrow}d_{A\vec{R}\downarrow}\right]$$

$$= -\frac{t}{2}\sum_{\langle\vec{R},\vec{R}'\rangle}\left\{\left[c^\dagger_{A\vec{R}\uparrow} + c_{A\vec{R}\downarrow}\right]\left[c_{B\vec{R}'\uparrow} - c^\dagger_{B\vec{R}'\downarrow}\right] - \left[c^\dagger_{B\vec{R}'\downarrow} + c_{B\vec{R}'\uparrow}\right]\left[c^\dagger_{A\vec{R}\uparrow} - c_{A\vec{R}\downarrow}\right]\right\}$$

$$= -t\sum_{\langle\vec{R},\vec{R}'\rangle}\left[c^\dagger_{A\vec{R}\uparrow}c_{B\vec{R}'\uparrow} + c^\dagger_{B\vec{R}'\downarrow}c_{A\vec{R}\downarrow}\right].$$ (A.12)

The other contribution to the hopping term can be shown to remain unchanged (in terms of the $c$'s) in a similar fashion.

Next we take up the BCS pairing term:

$$\Delta\sum_{\vec{R}}\left[d_{A\vec{R}\uparrow}d_{A\vec{R}\downarrow} + \text{H.c.}\right] = \frac{\Delta}{2}\sum_{\vec{R}}\left\{\left[c_{A\vec{R}\uparrow} + c^\dagger_{A\vec{R}\downarrow}\right]\left[c^\dagger_{A\vec{R}\uparrow} - c_{A\vec{R}\downarrow}\right] + \text{H.c.}\right\}$$

$$= -\Delta\sum_{\vec{R}}\left[c^\dagger_{A\vec{R}\uparrow}c_{A\vec{R}\uparrow} + c^\dagger_{A\vec{R}\downarrow}c^\dagger_{A\vec{R}\downarrow} - \frac{1}{2}\right].$$ (A.13)

However, the paring potential on the $B$ sublattice yields a potential term with the opposite sign:

$$\Delta\sum_{\vec{R}}\left[d_{B\vec{R}\uparrow}d_{B\vec{R}\downarrow} + \text{H.c.}\right] = -\frac{\Delta}{2}\sum_{\vec{R}}\left\{\left[c_{B\vec{R}\uparrow} - c^\dagger_{B\vec{R}\downarrow}\right]\left[c_{B\vec{R}\downarrow} + c^\dagger_{B\vec{R}\uparrow}\right] + \text{H.c.}\right\}$$

$$= \Delta\sum_{\vec{R}}\left[c^\dagger_{B\vec{R}\uparrow}c_{B\vec{R}\uparrow} + c^\dagger_{B\vec{R}\downarrow}c^\dagger_{B\vec{R}\downarrow} - \frac{1}{2}\right].$$ (A.14)

Adding the different contributions yields the following transformed Hamiltonian:

$$H'_c = H'_0 + H'_\Delta,$$ (A.15)

$$H'_0 = -t\sum_{\langle\vec{R},\vec{R}'\rangle,\sigma}\left[c^\dagger_{A\vec{R}\sigma}c_{B\vec{R}'\sigma} + 3\right],$$ (A.16)

$$H'_\Delta = -\Delta\sum_{\vec{R},\sigma}\left[c^\dagger_{A\vec{R}\sigma}c_{A\vec{R}\sigma} - c^\dagger_{B\vec{R}\sigma}c_{B\vec{R}\sigma}\right].$$ (A.17)

As pointed out at the beginning of this Appendix, this Hamiltonian describes a band insulator with a gap $\propto \Delta$. Furthermore, what makes it interesting from the point of view of this work is that, whilst the BCS pairing potential cannot be generalized from $SU(2)$ to $SU(N)$ without breaking this symmetry group, $H'$ admits a fully $SU(N)$ symmetric generalization. Another important point to stress is that from the superconductor to the insulator the mapping is possible as long as the initial model is particle-hole symmetric. Adding any particle-hole symmetry-breaking perturbation to $H$ in Eq. (A.3) will generate pairing potential terms in terms of the $c$'s. On the other hand, any term that is expressed in terms of local spin operators, e.g. $S^+_{p\vec{R}} = d^\dagger_{p\vec{R}\uparrow}d_{p\vec{R}\downarrow}$ ($p = A, B$ will take the same form:

$$S^+_{A\vec{R}} = d^\dagger_{A\vec{R}\uparrow}d_{A\vec{R}\downarrow} = \frac{1}{2}\left[c^\dagger_{A\vec{R}\uparrow} + c_{A\vec{R}\downarrow}\right]\left[c_{A\vec{R}\downarrow} - c^\dagger_{A\vec{R}\uparrow}\right] = c^\dagger_{A\vec{R}\uparrow}c_{A\vec{R}\downarrow},$$

and $S^-_{A\vec{R}} = \left[S^+_{A\vec{R}}\right]^\dagger = c^\dagger_{A\vec{R}\downarrow}c_{A\vec{R}\uparrow}$, and $S^z_{A\vec{R}} = \frac{1}{2}\left[S^+_{A\vec{R}}, S^-_{A\vec{R}}\right] = \frac{1}{2}\left[c^\dagger_{A\vec{R}\uparrow}c_{A\vec{R}\uparrow} - c^\dagger_{A\vec{R}\downarrow}c_{A\vec{R}\downarrow}\right]$, etc.

## B  $SU(N)$ pseudo-fermion representation of the impurity spin

As a reminder, we recall that the $SU(N)$ (i.e., special unitary) group is the group of $N \times N$ unitary matrices with determinant 1. The SU($N$) algebra is generated by $N^2$ generators $S^{\alpha\beta}$ which are represented as traceless hermitian matrices (i.e., tr $\{S^{\alpha\beta}\} = 0$), and which satisfy the commutation relation [38]

$$\left[S^{\alpha\beta}, S^{\gamma\delta}\right] = \delta_{\beta\gamma} S^{\alpha\delta} - \delta_{\alpha\delta} S^{\beta\gamma}. \tag{B.1}$$

The operator $\hat{N} = \sum_{\alpha=1}^{N} S^{\alpha\alpha}$ satisfies the property $\left[\hat{N}, S^{\alpha\beta}\right] = 0$, which implies that the number of independent generators is actually $N^2 - 1$. In particular, note that for $N = 2$, we recover the usual algebra of the SU(2) group, with the three independent generators $S^+ = S^{12}, S^- = S^{21}, S^z = \frac{1}{2}(S^{11} - S^{22})$ of the impurity spin.

We next introduce a representation of the SU($N$) generators $S^{\alpha\beta}$ in terms of (pseudo-)fermionic operators $f_\alpha$,

$$S^{\alpha\beta} \equiv f_\alpha^\dagger f_\beta - q\delta_{\alpha\beta}, \tag{B.2}$$

which are subject to the occupation constraint

$$\sum_{\alpha=1}^{N} f_\alpha^\dagger f_\alpha = qN. \tag{B.3}$$

Here $q$ is the $f$-electron filling factor controlling the total pseudo-fermion conserved charge $Q = qN = 1$, and setting the population of the $f$ electrons in different physical situations. In our case, where the physical situation corresponds to a SU(2) model, $q$ must be chosen as $q = 1/2$. However, it takes the more generic value $q = 1/N_j$, where $N_j = 2j + 1$ is the degeneracy of a multiplet of the total angular momentum $\vec{J} = \vec{L} + \vec{S}$ in an impurity orbital.

## C  Finding the saddle point

In this Appendix we provide details on the calculation of the saddle-point Eqs. (26) and (27). First, to solve Eq. (26), we compute the sum over Matsubara frequencies turning into the following integral in a contour $C$ on the complex frequency $z$ plane:

$$\sum_{i\nu_n} \frac{\mathcal{G}_{cc}^{(0)}(i\nu_n)}{i\nu_n - |V|^2 \mathcal{G}_{cc}^{(0)}(i\nu_n) - \lambda} = \frac{1}{2\pi i} \int_C dz \, \frac{\mathcal{G}_{cc}^{(0)}(z)}{z - |V|^2 \mathcal{G}_{cc}^{(0)}(z) - \lambda} \frac{-\beta}{1 + e^{\beta z}}$$

$$= \frac{-\beta}{\pi} \int_{-\infty}^{\infty} dz \, \text{Im}\left[\frac{\mathcal{G}_{cc}^{(0)}(z + i0^+) n_F(z)}{z - |V|^2 \mathcal{G}_{cc}^{(0)}(z + i0^+) - \lambda}\right]. \tag{C.1}$$

The contour $C = C_+ + C_-$ is made of $C_+$, which consists of the upper semi-circle and the straight segment $-\infty + i0^+ \to \infty + i0^+$ and $C_-$, which consists of the lower semi-circle and the segment $\infty - i0^+ \to -\infty - i0^+$. In the above expression and in what follows $n_F(z) = 1/(1 + e^{\beta z})$ denotes the Fermi-Dirac distribution and $0^+$ a positive infinitesimal. The choice of $C$ avoids the branch cut along the real $z$ axis. The integrals along the upper and lower semi-circles in $C_\pm$ vanish when their radius is taken to infinity. Therefore, the contour integral over $C$ only receives contributions from the straight segments $-\infty + i0^+ \to \infty + i0^+$ and $\infty - i0^+ \to -\infty - i0^+$.

The Matsubara sum in Eq. (27) can be calculated in the same fashion:

$$\sum_{i\nu_n} \frac{1}{i\nu_n - |V|^2 \mathcal{G}_{cc}^{(0)}(i\nu_n) - \lambda} = \frac{1}{2\pi i} \int_C dz \frac{1}{\frac{1-e^{-0^+ z}}{0^+} - |V|^2 \mathcal{G}_{cc}^{(0)}(z) - \lambda} \frac{-\beta}{1 + e^{\beta z}}$$

$$= \frac{-\beta}{\pi} \int_{-\infty}^{\infty} dz \operatorname{Im}\left[ \frac{n_F(z)}{\frac{1-e^{-0^+ z}}{0^+} - |V|^2 \mathcal{G}_{cc}^{(0)}(z + i0^+) - \lambda} \right], \quad \text{(C.2)}$$

where $\frac{1-e^{-0^+ z}}{0^+}$ in the denominator appears in the discrete version of the coherent-state path integral to ensure convergence of the functional integral [30].

The above Matsubara sums yield the following equations for self-consistency at finite temperature:

$$S_1(\lambda, |V|^2) = 0, \quad \text{(C.3)}$$

$$S_2(\lambda, |V|^2) = 0, \quad \text{(C.4)}$$

where,

$$S_1(\lambda, |V|^2) = \frac{1}{\pi} \int_{-\infty}^{\infty} dz \operatorname{Im}\left[ \frac{\mathcal{G}_{cc}^{(0)}(z + i0^+) n_F(z)}{z - |V|^2 \mathcal{G}_{cc}^{(0)}(z + i0^+) - \lambda} \right] - \frac{1}{J}, \quad \text{(C.5)}$$

$$S_2(\lambda, |V|^2) = \frac{1}{\pi} \int_{-\infty}^{\infty} dz \operatorname{Im}\left[ \frac{n_F(z)}{\frac{1-e^{-0^+ z}}{0^+} - |V|^2 \mathcal{G}_{cc}^{(0)}(z + i0^+) - \lambda} \right] + q. \quad \text{(C.6)}$$

We solve Eqs. (C.5) and (C.6) using Newton-Raphson method. To this end, we need to calculate the Hessian matrix of the free energy, i.e.

$$\mathcal{J}(\lambda, |V|^2) = \begin{pmatrix} \partial_\lambda S_1(\lambda, |V|^2) & \partial_{|V|^2} S_1(\lambda, |V|^2) \\ \partial_\lambda S_2(\lambda, |V|^2) & \partial_{|V|^2} S_2(\lambda, |V|^2) \end{pmatrix}. \quad \text{(C.7)}$$

Starting from initial values, $(\lambda_1, |V_1|^2)$, we update them according to the following equation:

$$\begin{pmatrix} \lambda_{N+1} \\ |V_{N+1}|^2 \end{pmatrix} = \begin{pmatrix} \lambda_N \\ |V_N|^2 \end{pmatrix} - s \mathcal{J}^{-1}(\lambda_N, |V_N|^2) \begin{pmatrix} S_1(\lambda_N, |V_N|^2) \\ S_2(\lambda_N, |V_N|^2) \end{pmatrix}, \quad \text{(C.8)}$$

where $s > 0$ is a numerical parameter used to control the stability of the convergence.

## D   Definitions of spectral weight

The spectral weights are defined from the Lehmann representation of the $T$-matrix in Eq. (42),

$$A_\sigma(\omega) = \frac{-1}{\pi} \operatorname{Im}\left[ \mathcal{T}_\sigma(\omega^+) \right] \quad \text{(D.1)}$$

$$= \frac{-1}{\pi} \operatorname{Im}\left[ \sum_{mn} \frac{e^{-\beta E_m} + e^{-\beta E_n}}{Z} \frac{|\langle m|\mathcal{O}_\sigma|n\rangle|^2}{\omega^+ + E_m - E_n} \right] \quad \text{(D.2)}$$

$$= \frac{-1}{\pi} \operatorname{Im}\left[ \sum_{mn} \frac{W_{\sigma;mn}}{\omega^+ + E_m - E_n} \right] \quad \text{(D.3)}$$

$$= \sum_{mn} W_{\sigma;mn} \delta(\omega - E_n + E_m) \quad \text{(D.4)}$$

$$= \sum_\epsilon W_\sigma(\epsilon) \delta(\omega - \epsilon), \quad \text{(D.5)}$$

where $|m\rangle$ and $E_m$ are the eigenstates and the corresponding eigenvalues. $Z = \sum_m e^{-\beta E_m}$ is the partition function. The spectral weight $W_\sigma^{\text{NRG}}$ is computed from the matrix elements of $\mathcal{O}_\sigma$ between the NRG eigenstates using the full-density-matrix scheme [53]. For the large-N theory, the spectral weights are calculated from,

$$W_\sigma^{\text{large-N}}(\epsilon) = \lim_{z \to \epsilon^+} \text{Re}\left[(z - \epsilon)\mathcal{T}_\sigma(z)\right].\tag{D.6}$$

# E Numerical renormalization group

Following Wilson [20], a logarithmic discretization of the tridiagonalized version of the Hamiltonian in Eq. (1) is carried out. We used the adaptive scheme in Ref. [54] with discretization parameter $\Lambda = 2$ for a constant density of states $\rho_0 = 1/2D$ of the normal state (i.e. for $\Delta = 0$). This results in the following Hamiltonian for a Wilson chain of $L$ sites:

$$H = \sum_{j=0}^{L-1} t_j \left[ f_{j\sigma}f_{j+1,\sigma}^\dagger + f_{j+1,\sigma}f_{j\sigma}^\dagger \right] + \Delta\left[f_{j\uparrow}f_{j\downarrow} + f_{j\uparrow}^\dagger f_{j\downarrow}^\dagger\right] + J\vec{S}\cdot\vec{s}_0,\tag{E.1}$$

where the hopping amplitude $t_j$ decays exponentially as $\sim \Lambda^{-j/2}$. In order to make the numerical computation more efficient, we use the method described in the main text and in Appendix A and apply the Bogoliubov transformation given in Eqs. (7),(10) to map the host Hamiltonian to an insulator Hamiltonian [37]:

$$H = \sum_{j=0}^{L-1} \left\{ t_j \sum_\sigma \left(c_{j+1,\sigma}^\dagger c_{j\sigma} + \text{H.c.}\right) + \Delta(-1)^j Q_{Z,j} \right\} + J\vec{S}\cdot\vec{s}(0).\tag{E.2}$$

The NRG is performed using the conserved U(1) quantum numbers $Q_Z$ and SU(2) spin quantum numbers $S_z, \vec{S}^2$, where

$$Q_Z = \sum_i Q_{Z,j} = \sum_{j=0}^{L-1}\left[n_{j,\uparrow} + n_{j,\downarrow} - 1\right],\tag{E.3}$$

$$\vec{S} = \vec{S}^{\text{imp}} + \sum_{j=0}^{L-1}\vec{S}_j.\tag{E.4}$$

We set the temperature $T \ll \Delta$ as an effective zero-temperature limit. The truncation of states happens at an energy scale $\omega \gtrsim 10\,\omega_j = 10\Lambda^{(1-j)/2}$ and we retain at least 1024 states in each iteration. Due to the presence of gap, the NRG computation is stopped at iterations of energy scale $\sim 10^{-5}\Delta \ll \Delta$ [55]. The spectral weights, $W_\sigma(\epsilon)$, are defined using the $T$-matrix [47] derived from the commutator $\mathcal{O}_\sigma = [d_{0\sigma}, H_{\text{imp}}]$. We note that $d_{0\sigma}$ is the operator in the original $d$-fermion basis. The spectral weight is defined from the Lehmann representation(c.f. Eq. (D.5)) using the NRG eigenstates. We make use of the full-density-matrix scheme [53] to obtain the spectral weights and broaden the data using a hybrid kernel. The spectral function, $A_\sigma(\omega)$, reads

$$A_\sigma(\omega) = \sum_\epsilon W_\sigma(\epsilon)\left[\Theta(\epsilon - \Delta)lG(\omega, \epsilon, a) + \Theta(\Delta - \epsilon)G(\omega, \epsilon, b)\right],\tag{E.5}$$

where the functions $lG(\omega, \epsilon, a)$ and $G(\omega, \epsilon, b)$ are defined as follows:

$$lG(\omega, \epsilon, a) = \frac{\Theta(\omega\epsilon)}{a|\omega|\sqrt{\pi}}\text{Exp}\left[-\left(\frac{\ln(|\omega|) - \ln(|\epsilon|)}{a} - a/4\right)^2\right],\tag{E.6}$$

$$G(\omega, \epsilon, b) = \frac{1}{b\sqrt{\pi}}\text{Exp}\left[-\left(\frac{\epsilon - \omega}{b}\right)^2\right].\tag{E.7}$$

Inside the BCS gap, the spectral peaks are broaden using a Gaussian kernel of width $b = \Delta/1000$. Outside the gap, the peaks of the continuum part of the single-particle spectrum are broadened using a Log-Gaussian kernel with a rather narrow broadening parameter, $a = 0.05$, on a logarithmic mesh binning $\sim 500$ points per decade with respect to the gap. Furthermore, the spectral functions are averaged with 64 twist parameters [56] taken from the interval $[1/64, 1]$.

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
