# Peer review of "A Large-$N$ Approach to Magnetic Impurities in Superconductors"

_SciPost Physics, doi:SciPost Phys. 18, 087 (2025)_

## Round 1 · Referee Report · Anonymous (Referee 1) · 2024-10-20

Report

The authors present a large-N mean field approach to the Kondo model for a magnetic impurity in a superconductor. They show results for the subgap states position and the local density of states which compare reasonably well with NRG calculations in the limit of large Kondo temperature.

In my opinion the approach and the results presented in this manuscript are sound and deserve to be published in Scipost. The manuscript is written in a rather pedagogical way, which I find very valuable.

The only point which I find a bit obscure is the choice of the band-width for the comparison of large-N and NRG calculations. As indicated in Eq. (25), the authors require that the density of states at the Fermi level, \rho_0, should be the same in
both calculations, i.e. \rho_0 = 1/(\pi t) = 1/2D, where t is the hopping element in the effective TB model and D is the NRG band-width. Thus, I would have expected that for the comparison they choose a ratio D/t such that \rho_0\Delta is the same in both calculations. However, taking the values indicated in Fig. 4(a) I find \rho_0\Delta= 0.0005 in the NRG calculation, and \rho_0\Delta = 0.005/\pi for the large-N one. There might be some misunderstanding from my side but in any case it would be worth that the authors could clarify this issue in their manuscript.

The authors might also consider giving a reference to some recent STM experimental works where YSR were detected and correlated with the Kondo effect in the normal state (see Communications Physics 6, 214 (2023) and references there in).

Notice also a typo in the sentence "We speculate that this is feature...".

Recommendation

Publish (easily meets expectations and criteria for this Journal; among top 50%)

  • validity: high
  • significance: good
  • originality: high
  • clarity: high
  • formatting: excellent
  • grammar: excellent

Author:  Chen-How Huang  on 2025-01-16  [id 5130]

(in reply to Report 1 on 2024-10-20)

We thank the referee for his/her time and effort in reviewing our manuscript. We also appreciate the positive feedback on our work and the suggestion to publish it in SciPost.

Regarding the referee’s comment on the comparison between large-N and NRG calculations:
"The only point I find a bit obscure is the choice of the bandwidth for the comparison between the large-N and NRG calculations. As indicated in Eq. (25), the authors require that the density of states at the Fermi level, ρ0​, should be the same in both calculations, i.e., ρ0= \frac{1}{\pi t} = \frac{1}{2D}​, where t is the hopping element in the effective TB model and D is the NRG bandwidth. Thus, I would have expected that for the comparison, they choose a ratio D/t such that ρ0Δ is the same in both calculations. However, taking the values indicated in Fig. 4(a), I find ρ0/Δ=0.0005 in the NRG calculation, and ρ0/Δ=0.005/π for the large-N one. There may be some misunderstanding on my part, but in any case, it would be useful if the authors could clarify this issue in their manuscript."

We thank the referee for pointing out this apparent discrepancy. Indeed, the product of the density of states at the Fermi energy, ρ0~1/D (where D is the host bandwidth), and the superconducting gap, Δ, characterizes the superconducting host but it does not contain information about the impurity. The properties of the combined system of superconductor and impurity are scaling functions depending of the ratio of two characteristic energy scales, one being the superconducting gap of the clean superconductor Δ, and the other the Kondo temperature of the impurity in the normal state T_K. For instance, in the weak coupling limit (which is not the focus of our manuscript), the latter is well approximated by the well-known expression:
T_K=D*exp(−1/J∗ρ0)
where J is the exchange coupling of the impurity with the host. In general, for T_K << D, the Kondo temperature is a function of the dimensionless product ρ0 * J. Thus, in our comparison of the large-N and NRG results, the chains used for the two approaches must have the same density of states at the Fermi energy in the normal state. In the large-N method, it is more convenient to use a tight-binding model with constant hoping amplitude t resulting in a density of states given by the circle law. However, for NRG it is more convenient to use for a Wilson chain with exponentially decreasing hopping amplitude that approximates a band with constant density of states in the normal case. Assuming scaling in the parameter of Δ/T_K, we have compared the spectral properties of the impurity shown in Figs. 4a and 4b as well as Fig. 5 for the same values of the ratio Δ/T_K.

Regarding the existence of scaling, we are grateful that the referee has brought up to our attention the paper by Huang et al (Nat. Comm.), where an instance of what has been explained above is shown in their Fig. 4b. Indeed, the latter shows the scaling of the experimentally measured YSR energy with T_K/Δ. Some small deviations from the scaling are observed, which may be due to the fact that real impurities are not described by the Kondo model but by an Anderson model which is characterized by more parameters than just J or T_K. This does not apply to our calculations which have been carried out for the Kondo model for which we expect the universal scaling to be accurate. In passing, we also mention another study where some of the present authors recently found another instance where the universal scaling in terms of Δ/T_K appears when studying the YSR states induced by magnetic impurities in spin-split superconductors, see C. H. Huang et al, Phys. Rev. Res. 6, 033022 (2024).

In the revised manuscript, we have included an extended explanation of our parameter choices along the above lines together with a citation to the work of Huang et al. and to Phys. Rev. Res. 6, 033022 (2024), as examples of the universal scaling upon which were rely. See paragraph highlighted in red leading to Eq. 25, on page 5 of the revised manuscript (since it is not possible to upload the revised manuscript, we have included an snapshot of this page in the attachment).

Attachment:

---

## Round 1 · Referee Report · Anonymous (Referee 2) · 2024-10-29

Strengths

1-High level of detail, all steps of the derivation are very clear.
2-Method seems to work well in the singlet regime.

Weaknesses

1-Limitation to particle-hole symmetric cases.
2-Limitation to strong-coupling (singlet) regime.
3-Unclear whether the method can indeed be applied to more complex situations.

Report

This work shows that a large-N approach for solving the single-impurity Anderson model with a superconducting bath gives reasonably accurate results (as benchmarked against the NRG solution). Since this problem can be approximately solved with a number of simple techniques, e.g. exact diagonalisation in the "zero-bandwidth" limit on a two-site cluster and the refinements of this approach (such as the "surrogate model solver"), atomic limit solution and its various generalizations, or standard perturbation theory (at least for the singlet state), which all provide reasonable or even very good approximations, the addition of large-N solver to this list does not represent by itself a significant advance, especially because the approach is limited to the particle-hole symmetric and strong-coupling situation. The method is potentially applicable to more complex models, and if this were demonstrated by the authors, the significance of this work would be greatly improved. Therefore I would strongly suggest that the authors consider extending this manuscript with some generalizations of the model (e.g. to the two-lead situation with phase-bias, as in the Josephson-Anderson model, or the inclusion of the Zeeman term). If the method provides accurate description also in those situations, the potential impact of the work would be much bigger. Another possible improvement would be inclusion of fluctuations to go beyond the saddle-point approximation and perhaps capture the properties of the doublet state.

This being said, I actually like the manuscript, it is written very clearly and the presentation is very good, I also like the great level of detail. To the best of my ability to judge, it is technically correct. Therefore I would not object to this work being published even in its present form.

Recommendation

Publish (meets expectations and criteria for this Journal)

  • validity: top
  • significance: low
  • originality: good
  • clarity: top
  • formatting: excellent
  • grammar: excellent

Author:  Chen-How Huang  on 2025-01-16  [id 5129]

(in reply to Report 2 on 2024-10-29)
Category:
reply to objection

We thank the referee for her/his time and effort in reviewing our manuscript, as well as for the positive feedback and the suggestion to publish it in SciPost.

Regarding the main point raised by the referee, we would like to clarify that the primary goal of this work (which we hope is the first in a series) is to describe the method, benchmark its performance, and discuss its limitations, all in great detail. This alone has resulted in a manuscript of considerable length. We intentionally avoided introducing applications at this stage to maintain focus on the technical details and benchmarking of the results for the well-understood single-impurity problem. We believe that including applications would divert the attention from the core purpose as well as resulting in an extremely long manuscript.

We intend to explore applications in future works, especially for systems for which benchmarking using NRG or other numerically exact methods is not possible. Thus we believe this manuscript should serve as the main reference for the method itself.

As for the issue of the requirement of particle-hole symmetry, we acknowledge that it is a limitation of the method, as we already mentioned in the last paragraph of Sec. IV (Conclusions section, “One major limitation…”). However, we believe there is still a sufficiently broad class of models for which our method can provide useful physical insights. In this regard, it is worth comparing this limitation to the neglection of quantum fluctuations in the classical approach of Yu, Shiba, and Rusinov. Even If it is not obvious that one can neglect fluctuations, the classical approach has been widely and uncritically used to study many different models of impurities in superconductors. It took many years until the accuracy of the method was benchmarked using NRG and the conditions for its applicability were established (not only large impurity spin S, but also a large in magnitude and negative single-ion anisotropy, see R. Zitko, Physica B (2018) 536, 230-234). In our large-N approach, the benchmarking has been carried out in the present study and we hope that it can be widely applied with much more confidence despite being limited to systems with particle-hole symmetry.

---

## Round 2 · Referee Report · Anonymous (Referee 1) · 2025-1-22

Report

The authors have revised their manuscript in accordance to my previous report. I find that it is now suitable for publication.

Recommendation

Publish (easily meets expectations and criteria for this Journal; among top 50%)

---

## Round 2 · Referee Report · Anonymous (Referee 2) · 2025-2-3

Report

I recommend the publication of the manuscript in its present form.

(Although the presentation issues due to different densities of states in NRG and large-N approach could have been easily avoided by performing NRG calculations with the cosine DOS, which is not difficult to do.)

Recommendation

Publish (meets expectations and criteria for this Journal)

---

## Round 2 · Author Response

Dear SciPost Editor,
We thank you for forwarding the referee reports on our manuscript entitled "A Large-N Approach to Magnetic Impurities in Superconductors."
We are resubmitting a revised version of the manuscript with minor modifications, as suggested by the referees. We appreciate the positive feedback and constructive criticisms from both reviewers. Below we have addressed the points raised by them.
Given the ongoing effort in the Physics community to achieve a deeper understanding of the quantum phases of magnetic impurities in superconductors, we believe our paper is both timely and relevant to the field. We kindly ask that you consider it for publication in SciPost.
Sincerely,
Chen-How Huang, Alejandro M. Lobos, and Miguel A. Cazalilla

---

## Round 2 · List of Changes

In the paragraph between Eq.24 and Eq. 25, on page 5 of the revised manuscript. We have included an extended explanation of our parameter choices together with a citation to the work of Huang et al. and to Phys. Rev. Res. 6, 033022 (2024), as examples of the universal scaling.

---

## Editorial Decision

published